# The relationship between prenatal heat exposure and birth outcomes: How much does the heat metric matter?

**Mary-Alice Doyle**[1]*, **Bernard Leckning**[2]

1 Department of Social Policy, London School of Economics, London, United Kingdom, 2 Menzies School of Health Research, Charles Darwin University, Darwin, Australia

* m.s.doyle@lse.ac.uk

## Abstract

The impact of prenatal heat exposure on birth outcomes is well-established, but what is it about heat drives this relationship? Is it exposure to extreme temperatures, to moderate heat, or the confluence of heat and humidity? Despite the large body of research on heat exposure and birth outcomes, the literature lacks consistent measurement. This means we cannot extract practical recommendations around which heat conditions pose the greatest risk, and hence should be avoided during pregnancy. It also means we cannot predict the implications of climate change on neonatal health and healthcare needs at a population level. This paper has two goals: first, to demonstrate that our conclusions around the existence and magnitude of the impact of heat exposure vary dramatically with the choice of heat exposure metric, and second, to make general recommendations for how heat exposure should be measured in future. We present analysis from Australia's Northern Territory — a region spanning tropical and arid climates. We compare commonly used heat exposure metrics, alongside additional metrics supported by theory. We find that a metric based on 'bands' of exposure and incorporating daily minimum as well as maximum measures provides the best fit; this is consistent with our theoretical understanding that both moderate and extreme heat affect fetal development in different ways. Estimates based on our preferred heat metric suggest that the impact of prenatal heat exposure on preterm birth is orders of magnitude larger than what would be implied by some metrics commonly used in the literature. Our findings underscore the importance of getting the measure of heat right, particularly in tropical climates.

## 1. Introduction

There is a wealth of evidence that ambient heat exposure adversely affects human health [1,2]. The impact of prenatal heat exposure on a baby's health at birth is particularly well-studied. Chersich et al.'s [3] systematic review finds that heat exposure is

**Data availability statement:** The study used de-identified linked health data which is hosted on a secure cloud-based platform with restricted access. Access is restricted to Human Research Ethics Committee approved investigators and projects as per conditions set by data custodians to protect sensitive information. Applications for data access should be directed to the CYDRP Research Program Leader, Bernard Leckning: bernard.leckning@menzies.edu.au. Queries regarding ethics approval should be directed to ethics@menzies.edu.au.

**Funding:** This research was funded by a London School of Economics PhD Studentship. The funders had no role in study design, data collection and analysis, decision to publish, or preparation of the manuscript.

**Competing interests:** The authors have declared that no competing interests exist.

associated with a higher risk of preterm birth, lower average birthweight, and a higher risk of stillbirth. These effects may have long-term consequences: prenatal heat exposure has been linked to lower levels of physical health, mental health, education and earnings in adulthood [4–7].

But what exactly is the nature of the relationship between heat and newborns' health? Is fetal development damaged by short-lived exposure to extreme temperatures? Or should we be more concerned about prolonged exposure to moderately high temperatures? Or is it instead the interaction of heat with humidity that poses the greatest risk?

We are unable to answer these questions because, to date, research on this topic has lacked consistency in the way that heat exposure is defined and measured. But answers to these questions are important: in order to anticipate the likely impact of climate change on healthcare needs, and mitigate the impacts of heat exposure on population health, it is important to understand what level and duration of heat exposure leads to poorer health at birth.

In this paper, we analyse how much the way we measure heat exposure matters for our conclusions. To do this, we first set out options for measuring heat exposure, including the three measures of heat exposure that are most common in the literature, alongside two additional measures motivated by our conceptual framework: we call these our five 'heat metrics'. We estimate regressions with each of the five metrics, and combinations of multiple metrics together. We identify a preferred metric in our context, based on standard measures of goodness-of-fit. We then demonstrate how reliance on non-preferred metrics would affect our conclusions about the impact of prenatal heat exposure on birth outcomes.

In our analysis, we use data from the Northern Territory of Australia – a region spanning tropical and arid climates. The fact that we use data from these climate zones is important, because around half of the world's population lives in climates like these [8,9], yet most empirical research on this topic uses data from cooler, less humid climates (e.g., in Chersich et al.'s [3] systematic review, around one-third of studies relate to subtropical and semi-arid climates, but with just one from an arid climate and none from equatorial/tropical climate zones). Using data from the Northern Territory means we can analyse exposure to very high temperatures (e.g., above 35 degrees Celsius), and exposure to hot, humid conditions – which are either not experienced at all in other climates, or are experienced so rarely that their effects cannot be reliably estimated.

There is good reason to expect that the choice of heat metric is particularly important in tropical climates. In temperate climate zones, which cover much of the USA, Europe, and East Asia, correlations between various measures of heat exposure are high, meaning that one measure of heat exposure (e.g., maximum daily temperature) is a good proxy for others (e.g., minimum daily temperature, or wet bulb temperature). Fig 1 demonstrates this, showing the correlations between daily minimum and daily maximum temperatures in a handful of major cities – the correlations are high in cities with temperate climates.

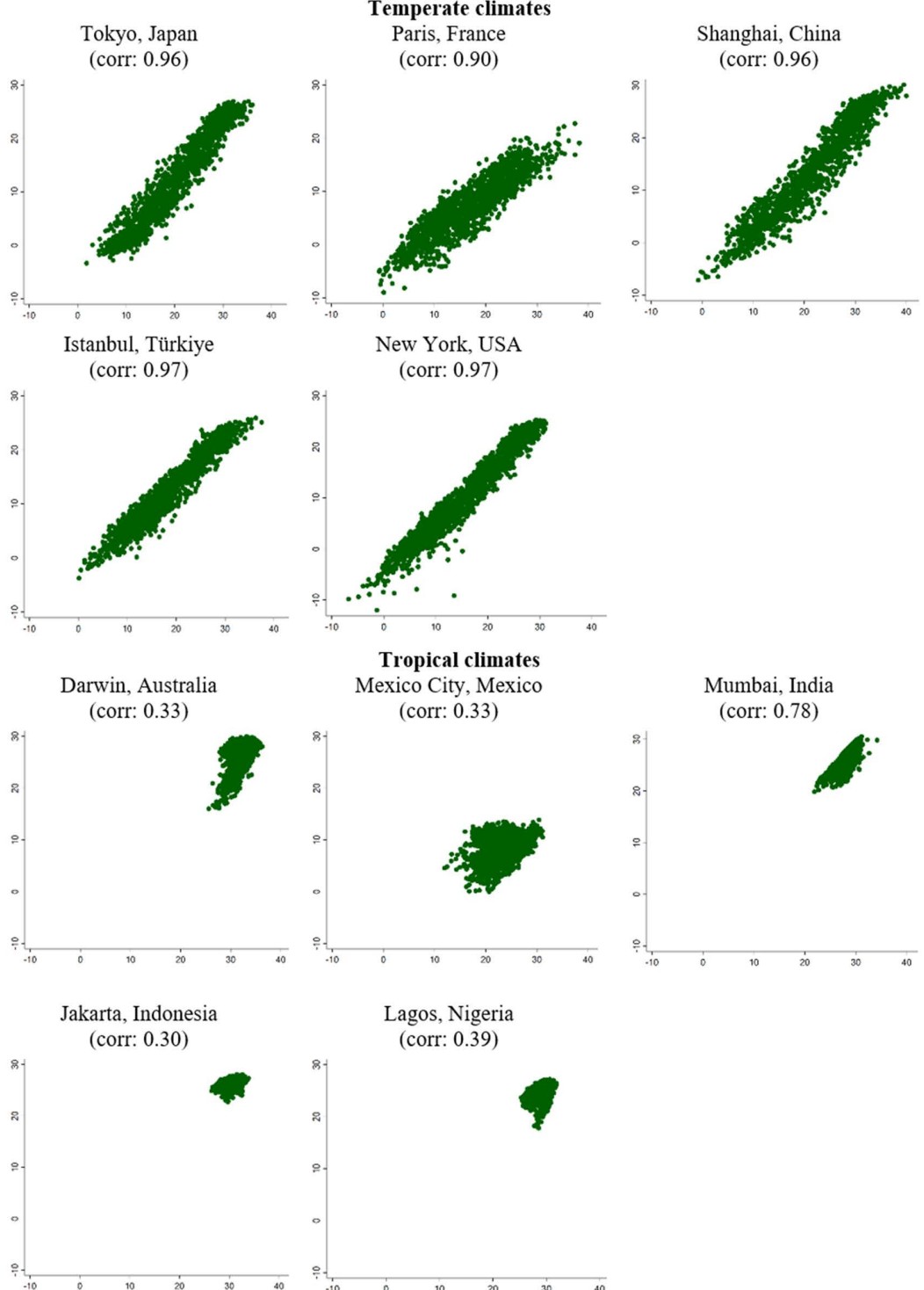

**Fig 1. Scatter plot of daily maximum and minimum temperature, 2020–2023.** Source: NASA reanalysis data.

However, in tropical and subtropical climates, this is not the case. For instance, in Lagos, Mexico City, Mumbai, Jakarta, and Darwin (the capital city of the Northern Territory), Fig 1 shows that there are relatively low correlations between maximum and minimum daily temperature. The same is true for the correlation between air temperature and wet bulb temperature (see S1 Fig). Correlations are substantially higher in the Northern Territory's arid climate zones, but still lower than in temperate zones shown in Fig 2 (see S2 Fig). In tropical climates, one measure is not a good proxy for all others, and therefore we must carefully consider which aspect of heat exposure we want to measure.

This paper makes two main contributions to the knowledge base on the link between prenatal heat exposure and birth outcomes.

First, we quantify how much the way we measure heat exposure matters. Given the large body of research that finds an impact of heat exposure on birth outcomes, we are confident that this causal relationship exists. But every recent review on this topic has highlighted a lack of consistency in choice of heat metric [3,10–14]. The problem with this inconsistency is that when different metrics are applied in different populations, we do not know whether it is the population, the metric, or something else entirely, that explains differences in findings. Our analysis compares alternative heat metrics within a single population. We find that both the existence and size of the estimated relationship between heat exposure and birth outcomes can vary depending on the metric chosen. For example, we find that a metric based on average maximum daily temperatures alone – as is common in the literature – captures less than half of the impact that our preferred metric estimates.

Second, we analyse which measurement choices matter most. Recently, some researchers have questioned the use of air temperature as a default in this literature [15–17]. Their reasoning is that metrics which measure both air temperature and humidity together (e.g., wet bulb temperatures and heat indices), will more accurately reflect people's experiences of heat than air temperature alone. However, we find that wet bulb temperatures have limited explanatory power in the population we study, and instead, other metrics – e.g., including both maximum and minimum daily temperatures – provider better explanatory power. Unlike wet bulb and heat indices, these measures are also more readily available, meaning there is little barrier to their use. Our discussion reflects on these and other findings, setting out some recommendations for researchers analysing the impact of prenatal heat exposure.

This paper also provides new evidence on how heat exposure affects birth outcomes in the Northern Territory of Australia. We estimate that typical seasonal variation in heat exposure contributes to a 4.5 percentage point higher risk of

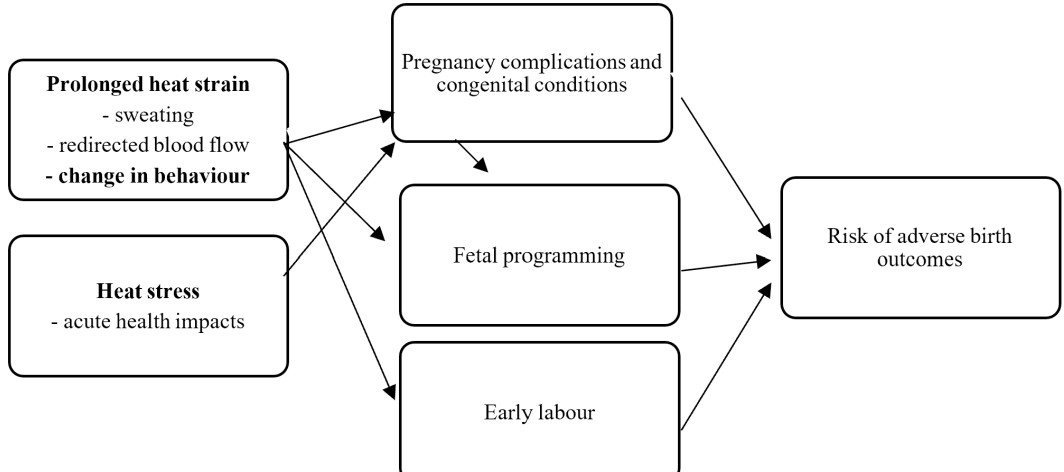

**Fig 2. Summary of mechanisms through which heat stress and heat strain may affect birth outcomes.** Source: Authors' analysis.

preterm birth at some times of year. This is large – for example, close in magnitude to the risks of frequent smoking during pregnancy [18].

In our analysis we focus on preterm birth, as this is the birth outcome most commonly studied in previous research. Preterm birth is associated with poorer health and developmental outcomes later in childhood [19] and leads directly to an increase in healthcare costs by triggering admission to neonatal intensive care units [20]. However, as we will discuss in our conceptual framework, heat exposure may affect fetal development in a range of ways, most of which do not necessarily lead to preterm birth. It is therefore important to note that we find impacts of heat exposure that are consistent across five different measures of health at birth; this confirms that heat exposure affects fetal development and hence health at birth in general, the effect is not isolated to a single outcome.

The rest of this paper proceeds as follows. Section 2 sets out a conceptual framework, outlining the possible causal pathways that may explain the impacts of heat exposure on prenatal development, and how they may be measured. Section 3 provides details on the Northern Territory, the administrative data we use, the heat exposure metrics we consider, and our estimation methods. Section 4 presents our data analysis, in which we identify our preferred heat metric and discuss conclusions we may draw if we instead used alternative heat metrics. Section 5 discusses practical implications of using non-preferred heat metrics, and recommendations for future analysis, before turning to our conclusion in Section 6.

## 2. Conceptual framework

### 2.1. What is heat exposure?

Heat exposure it often discussed interchangeably with air temperature – that is, the temperature that can be measured with a standard thermometer. However, while air temperature is a major contributor to the level of heat that a person experiences, there are additional contributors. As McGregor and Vanos [21] explain in their primer on the physiological impacts of heat on the human body, these include other weather conditions (humidity, windspeed, radiation), individual factors (levels of exertion, pre-existing medical conditions, and medications), and the built environment.

Many researchers argue that humidity is particularly important [15,17,22]. This is because one way that the body cools itself is through having sweat evaporated from the skin, and sweat evaporates more slowly when it is humid. Therefore, for a given temperature, higher humidity increases the body's risk of overheating.

In deciding how to measure heat exposure, we must understand which aspects – such as air temperatures, humidity or other contributors – matter for prenatal development. The difficulty here is that there are multiple mechanisms through which heat exposure during pregnancy can affect the mother and developing fetus [23], and hence lead to poorer birth outcomes. We do not yet know from physiological science which of these mechanisms is most important [24]. In the remainder of this section, we explain these mechanisms (summarised in Fig 2), before returning to the question of which measures may best capture these effects.

### 2.2. How does heat exposure affect the fetus?

We set out three channels through which heat exposure may affect prenatal development: heat stress, heat strain and maternal behaviour.

It is important to note that while these mechanisms can increase the risk of adverse birth outcomes like preterm birth – which is the focus of this study – they will not necessarily do so. They may affect fetal development in ways that impact a child's health and development in infancy and/or later in life, but which do not necessarily lead to preterm birth.

**Heat stress.** Research on the impact of heat exposure on health tends to focus on heat stress; that is, the acute health conditions that can result from an increase in the core body temperature [21]. These effects include heat stroke, cardiovascular stress, respiratory stress, and acute kidney failure [2,25]. Heat stress typically occurs when air temperatures are around or above the average body temperature (37 degrees C). However, it can occur at lower air temperatures [2], for instance due to high humidity or high levels of exertion.

 

We know from both animal studies and epidemiological evidence that maternal heat stress during pregnancy can cause birth defects [24,26]. This may happen either through the fetus itself overheating, or indirectly through the heat stress-induced condition that the mother is facing. Some suggestive evidence also indicates that heat stress early in pregnancy may affect the development of the placenta, increasing the likelihood of pre-eclampsia [27], and that it may affect neural tube formation, leading to neural tube defects [28,29].

**Heat strain.** Even when conditions are not hot enough to cause heat stress, heat may still impact fetal development because it causes 'heat strain' for the mother [21]. Heat strain involves the normal workings of the thermoregulatory system to cool the body, and hence avoid heat stress: that is, by sweating and by redirecting blood flow from the core of the body to the skin to dissipate heat [30].

If heat strain is experienced for a prolonged period, it may affect the fetus. This can happen because of sweat-induced dehydration. It can also happen because when the mother's blood flow is redirected to her skin, this means reduced blood flow to the placenta, and hence reduced flow of nutrients to the fetus [24]. Both channels have been evidenced in experimental animal studies – where researchers have been able to control heat exposure and fluid intake in a way that is not possible in epidemiological research. These studies show long-term 'fetal programming' effects on offspring health [31,32].

These potential fetal programming effects explain how the mother's natural responses to heat strain may reduce nutrient flow to the fetus – if this continues for a long period, it can affect the fetus' development. This could happen at any time during pregnancy. In addition, epidemiological evidence suggests that exposure to moderate heat, if experienced towards the end of pregnancy, can bring forward labour leading to increased rate of preterm birth [33].

**Maternal behaviour.** Beyond physiological effects, prolonged heat exposure can also affect maternal behaviour, with flow-on effects to the fetus. Heat exposure reduces metabolism and appetite [30], which may mean changed food and drink consumption patterns [34]. Heat may also reduce quality and length of sleep [35] and lead to changes in exercise patterns [24].

### 2.3. Implications for heat exposure measurement

Given the mechanisms outlined above, we draw three takeaways for how we might construct heat exposure metrics.

First, it is not obvious which underlying measure of heat we should use. Within the literature, heat exposure is almost universally measured based on air temperature. But as described above, a person's experience of heat is a result of a range of factors, including temperature, humidity, wind and radiation. Therefore, in theory, a data series incorporating these variables may provide a more accurate picture of true heat exposure. One option is wet bulb temperature, which can be observed directly, and reflects air temperature and humidity – other options include Apparent Temperature, Heat Index, Universal Thermal Climate Index, and Wet Bulb Global Temperature [15]. However the reason that this is not an obvious choice is that, empirically, we do not see a consistent link between high humidity and poorer health outcomes. As Baldwin et al. [16] suggest, there may be a range of explanations for this, including the fact that a build-up of humidity when it is not raining has different effects than high humidity while raining.

Second, that the effects of extreme heat – that is, heat at levels around or above normal core body temperatures – may be different from the effects of moderate heat, because extreme heat is more likely to cause heat stress. This means estimates of the impact of heat exposure based on average temperatures over a given period are likely mis-specified, as an average will fail to distinguish between conditions likely to lead to heat strain vs heat stress.

Third, that even moderate levels of heat, when experienced for a prolonged period, may affect fetal development by causing heat strain or changes in maternal behaviour. If this is the case, then it may be important to choose metrics that reflect prolonged heat exposure – such as high daily minimums, reflecting at least 24 hours of high temperatures – in addition to the metrics used in the overwhelming majority of studies which measure peak levels of heat exposure (i.e., high daily maximums). This may be particularly important in tropical climates where daily maximums and minimums are not strongly correlated with one another (as shown in Fig 1).

In our analysis, we estimate the impact of heat exposure on preterm birth, using a range of different heat metrics, to test which fits the data best. In line with this conceptual framework, we test metrics that include both air temperature and wet bulb temperature, metrics based on daily maximums and minimums, and metrics allowing for non-linear effects of heat exposure. We compare these metrics with those most commonly used in prior research (described below). However, our conceptual framework also points out a potential moderator between heat exposure and birth outcomes that we are unable to measure in our data – maternal behaviour. While maternal behavioural responses to heat exposure do not feature in our data analysis, behaviour is likely a key factor, and may help to explain differences in the magnitudes and timing of effects (in terms of whether exposure has different effects depending on the timing of exposure during pregnancy) across different populations and contexts.

## 3. Materials and methods

### 3.1. Study context

The Northern Territory (NT) is one of Australia's eight states and territories. The NT is a large region covering the central part of northern Australia, in which residents face high levels of heat exposure. The NT has a population of around 233,000, 60 percent of whom live in or around the capital city of Darwin [36]. Previous research shows that heat exposure plays a major role in explaining month-to-month variation in average birth outcomes in the NT [37].

The tropical north of the NT, where Darwin is located, is hot and humid. Temperatures vary within season but, on an average day in the wet season (November to April), temperatures range between 25–33 degrees Celsius, and in the dry season (May to October), between 20–30 degrees. The hottest time of year is October-December, in the 'build up' to the wet season – when temperatures and humidity are high, and there is little rainfall. Heavy rainfall usually begins in late December, though the timing can vary from year to year.

The central and southern parts of the NT have an arid climate, with very hot summers and mild or cold winters. In Alice Springs, the largest town in the region, temperatures on an average summer day range between 20 and 35 degrees, and on an average winter day, between 4 and 20 degrees.

Around one-third of the NT population identify as Aboriginal. There are significant differences between the Aboriginal and non-Aboriginal populations in the NT, in terms of geography, heat exposure and economic resources. Eighty percent of Aboriginal residents in the NT live outside of Darwin, many in remote Aboriginal communities which experience more extreme weather conditions. In remote communities, many houses are poorly insulated, and many residents face energy poverty [38].

### 3.2. Data

We combine administrative birth records with daily weather data.

Our analysis sample includes 95 percent of all babies who were born in the NT and conceived between March 2000 and September 2009, a sample of 34,258 children. There were 35,899 babies born during this period. Our sample excludes 1,020 births for which the mother's place of residence could not be geocoded and therefore could not be matched to weather data, and 621 births for which some covariates were missing. We define our sample based on date of conception instead of date of birth because definitions based on date of birth will systematically exclude children born preterm at the beginning of the sample period, and exclude those born late term at the end of the sample period – this could lead to bias, especially when analysing the impact of seasonal exposures like heat [39]. We determine date of conception by subtracting gestational age (in weeks) from birthdate.

We include only births to mothers whose usual place of residence is in the NT, and could be geo-coded. There are a small number of births for which the mother's place of residence as entered in the perinatal data could not be found, either using a fuzzy match with the R package 'geonames', or through manual search on Google, the NT Place Names Register

(https://www.ntlis.nt.gov.au/placenames/) and BushTel (https://bushtel.nt.gov.au/). We include stillbirths (making up under 1% of births) and plural births (under 2.5% of births) in the analysis sample.

The birth record data were extracted from administrative records and prepared for analysis between 3 July and 31 August 2017. Data analysis for research purposes for this paper was conducted between 26 June 2023 and 19 December 2024 using Stata 17. Analysis data are de-identified, and the authors do not have access to information that could identify individual participants during or after the data were collected. Approval for the research has been provided by the Northern Territory Department of Health and Menzies School of Health Research Human Research Ethics Committee (Ref: 18–3261). The data linkage project's First Nations Advisory Group have independently reviewed and endorsed the research to ensure it is respectful of Aboriginal perspectives.

We link these records to NASA's daily weather reanalysis data, based on the mother's place of usual residence at the time of the birth. We do not have data on place of residence throughout pregnancy, therefore we assume that the place of residence at birth corresponds with the place the mother spent most of her time during pregnancy. The weather data are available at intervals of 0.5 x 0.625 degrees of latitude and longitude (roughly 50x55km). This gives a total of just over 500 cells throughout the NT, 166 of which have births within our analysis period – an average of 203 births per cell. We link data based on date of conception, such that the first trimester is the first 12 weeks from the conception date, the second trimester is the following 14 weeks, and the third trimester is the following 13 weeks. Definitions for timing and length of trimesters vary across contexts, but these are the definitions commonly used in Australia. We analyse 39 weeks of gestation, as this is the average gestational length in our population.

The weather reanalysis data comes from NASA's model using ground station and satellite data. Some may be concerned about accuracy of these data compared with traditional weather station observations. However, in our context, the NASA data lines up very closely with observations from the Australian Bureau of Meteorology's weather stations (see S1 Appendix for details). We use the reanalysis data because they are highly localised and have no missing observations – which is not the case for weather stations across remote parts of the NT. For instance, over this period some weather stations are decommissioned, and others have systematic missing observations (i.e., in the wet season when they become inaccessible due to flooding and therefore observations are systematically missing during periods of extreme weather).

### 3.3. Heat exposure metrics

We construct five different heat exposure metrics. The first three are based on common metrics used in prior research, while the next two build on the conceptual framework outlined above. Our metrics are:

**Benchmark.** This is a count of the number of days in each trimester with maximum daily temperatures within 5-degree ranges (under 20, 20–24.99, 25–29.99, 30–34.99, 35–39.99 and 40+). In our analysis, the omitted category is 25–29.99 degrees. This metric allows for a flexible functional form, and it is the approach that Dell et al. [40] recommend in cases when researchers are agnostic about how the heat metric should be specified. It has been widely used [4,5,33,41–43].

We take this as our 'benchmark' metric, because, out of the metrics that are most commonly used in the literature, this best allows for the non-linearities that we expect to see, based on our conceptual framework.

**Average temperatures.** This is a simple average of maximum daily temperatures in each trimester in pregnancy. This metric is also widely used [44–49]. It assumes linear effects of each additional degree, and hence does not allow for different effects for higher temperatures most likely to cause heat stress.

**Heatwave count.** This is a count of the number of heatwaves in each trimester of pregnancy. We calculate this based on the Excess Heat Factor, which measures the extent to which daily air temperatures are unusually high for a given location and time of year [50]. It is calculated by combining a) average daily temperatures in the previous 3 days, compared with the past 30 days, and b) average daily temperatures in the previous 3 days compared with the long-term location average. A heatwave is defined as three or more consecutive days with a positive Excess Heat Factor. Full

details of how this is constructed are available in Nairn and Fawcett [50]. We use this measure as it is the one used by the Australian Bureau of Meteorology.

Counts of extreme heat events (whether defined as heatwaves or not) are common in the literature [51–55]. Such a metric allows for nonlinearities in the effects of heat exposure, but imposes a specific structure on that nonlinearity; it assumes a threshold over which effects occur, and that there is no impact of exposure to more moderate heat conditions below that threshold.

**Max and min.** This is an enhanced version of the benchmark metric, where, in addition to maximum temperatures (under 20, 20–24.99, 25–29.99, 30–34.99, 35–39.99 and 40+), we include counts of the number of days in each trimester with minimum air temperatures within 5-degree ranges (under 5, 5–9.99, 10–14.99, 15–19.99, 20–24.99, 25+). In our analysis, the omitted category for minimum temperatures is 15–19.99. This metric reflects the implication from our Conceptual framework that daily minimum temperatures – which, if high, reflect prolonged exposure to heat – may indicate higher risk of heat strain continuing for long enough to affect the developing fetus.

**Wet bulb.** This metric is analogous to the 'Max and min' metric described above, but using wet bulb temperatures as the underlying data series, instead of air temperature. We construct counts of the number of days within each trimester with daily average wet bulb temperatures with ranges of <10, 10–14.99, 15–19.99, 20–24.99, 25+, and the number of days with daily maximum wet bulb temperatures within ranges of <10, 10–14.99, 15–19.99, 20–24.99, 25–29.99 and 30+. This metric reflects the possibility highlighted in our Conceptual framework that wet bulb temperatures may measure heat exposure more directly than air temperatures.

Maximum wet bulb temperature is not available directly within the NASA data, but we construct this measure using daily maximum combined with relative humidity, using Stull's equation [56]. We use average wet bulb temperature instead of minimum because this measure is directly available. However, as daily average is a linear transformation between maximum and minimum, the relationship between maximum and minimum (for air temperature), and maximum and average (for wet bulb temperature) should be constant.

Table 1 sets out the mean, standard deviation, maximum and minimum of these measures over the full pregnancy, for babies in our sample. During an average pregnancy, there are 2.4 heatwaves and just over one week with maximum temperatures above 40 degrees – but this varies greatly, with some pregnancies experiencing up to 9 heatwaves, and 98 days with maximum temperatures above 40 degrees.

### 3.4. Outcome measure

Our main outcome measure is preterm birth, defined as birth before 37 complete weeks of pregnancy. In our population, 10 percent of babies are born preterm (Table 1). We focus on preterm birth for comparability with previous research: it is the most commonly studied birth outcome. However, our conclusions hold across four additional measures of health at birth: birthweight, small for gestational age, Apgar scores, and admission to a special care nursery. Estimates using these outcomes are presented in the supplementary material.

Preterm birth is an important intermediary outcome: a large body of research tells us that children born preterm are more likely to face worse health and educational outcomes [19,57,58]. In the NT, children who were born preterm are more likely to be assessed as developmentally vulnerable at age 5 [59].

Recent research demonstrates that this relationship is not fixed: the predictive power of preterm birth has declined over time [60], and many children born preterm may face no detectable long-term effects [61]. This makes sense, for two reasons. First, advances in neonatal healthcare have greatly improved outcomes for preterm-born infants [62]. Second, because preterm birth has many causes [63]. The aetiology of preterm birth, and hence the long-term outcomes associated with it, may differ over time and across contexts. However, even in the desirable situation where preterm birth does not increase the risk of long-term developmental vulnerabilities for an individual, it remains an outcome of interest given the high cost of providing remedial neonatal care [20].

**Table 1. Descriptive statistics.**

| | Mean | St. deviation | Min. | Max. |
|---|---|---|---|---|
| **Birth outcomes** | | | | |
| Preterm birth (probability) | 0.10 | 0.31 | 0.00 | 1.00 |
| Birth weight (grams) | 3282.29 | 532.71 | 2160.00 | 4180.00 |
| SGA (probability) | 0.14 | 0.35 | 0.00 | 1.00 |
| Apgar 5 score | 8.89 | 1.22 | 0.00 | 10.00 |
| Special care nursery (probability) | 0.18 | 0.39 | 0.00 | 1.00 |
| **Heat exposure in 39 weeks to birth** | | | | |
| Maximum temp (average) | 31.23 | 1.74 | 24.60 | 37.84 |
| Minimum temp (average) | 22.43 | 4.21 | 10.33 | 28.03 |
| Daily average wet bulb (average) | 21.67 | 4.57 | 9.47 | 26.41 |
| Maximum wet bulb (average) | 24.43 | 3.84 | 11.72 | 30.00 |
| Number of heatwaves | 2.36 | 1.80 | 0.00 | 9.00 |
| Excess Heat Factor (average) | −9.31 | 10.45 | −46.29 | −0.60 |
| Daily maximum temperature bands | | | | |
| Days max temp <15 | 0.41 | 1.68 | 0.00 | 21.00 |
| Days max temp 15–20 | 4.53 | 11.18 | 0.00 | 64.00 |
| Days max temp 20–25 | 10.17 | 18.45 | 0.00 | 83.00 |
| Days max temp 25–30 | 85.75 | 47.81 | 0.00 | 266.00 |
| Days max temp 30–35 | 129.41 | 51.03 | 6.00 | 253.00 |
| Days max temp 35–40 | 34.78 | 36.90 | 0.00 | 161.00 |
| Days max temp 40+ | 7.96 | 14.02 | 0.00 | 98.00 |
| Daily minimum temperature bands | | | | |
| Days min temp<5 | 5.19 | 12.74 | 0.00 | 59.00 |
| Days min temp 5–10 | 10.83 | 21.12 | 0.00 | 90.00 |
| Days min temp 10–15 | 16.30 | 23.12 | 0.00 | 87.00 |
| Days min temp 15–20 | 34.88 | 28.83 | 0.00 | 128.00 |
| Days min temp 20–25 | 87.65 | 33.28 | 3.00 | 225.00 |
| Days min temp 25+ | 119.15 | 76.18 | 0.00 | 271.00 |
| Daily average wet bulb temperature bands | | | | |
| Days avg wet bulb <10 | 21.14 | 40.33 | 0.00 | 168.00 |
| Days avg wet bulb 10–15 | 19.42 | 27.37 | 0.00 | 122.00 |
| Days avg wet bulb 15–20 | 36.07 | 23.44 | 0.00 | 121.00 |
| Days avg wet bulb 20–25 | 81.62 | 32.12 | 2.00 | 204.00 |
| Days avg wet bulb 25+ | 114.75 | 70.02 | 0.00 | 247.00 |
| Daily maximum wet bulb temperature bands | | | | |
| Days max wet bulb <10 | 0.78 | 2.57 | 0.00 | 17.00 |
| Days max wet bulb 10–15 | 13.90 | 28.27 | 0.00 | 114.00 |
| Days max wet bulb 15–20 | 25.86 | 35.25 | 0.00 | 142.00 |
| Days max wet bulb 20–25 | 65.24 | 28.50 | 0.00 | 167.00 |
| Days max wet bulb 25–30 | 163.61 | 73.37 | 0.00 | 272.00 |
| Days max wet bulb 30+ | 3.61 | 8.29 | 0.00 | 53.00 |

Birthweight variable is top- and bottom-coded at the 2.5th and 97.5th percentiles, to reduce the influence of extreme outliers. SGA is a binary indicator, equal to 1 if birthweight is below the national 10th percentile for a given gestational age and sex (Dobbins et al. 2012). Apgar 5 is an index with values from 0 to 10, based on the birth attendant's judgement of the newborn's skin colour, heart rate, reflex, muscle tone and respiratory effort, 5 minutes after birth.

Source: NASA and Analysis dataset, average birth outcomes and heat exposure for all babies conceived in the NT from March 2000 to September 2009.

## 3.5. Analytical methods

Our goal in this analysis is to assess whether and how our estimates of the causal impact of heat exposure on preterm birth change when we use different heat exposure metrics. A challenge in doing so is that heat exposure may be correlated with birth outcomes for reasons which do not necessarily reflect the causal impact of heat itself.

Of primary concern is omitted variable bias. Heat exposure varies both over time, and across locations. But most of the variation in birth outcomes over time and across locations is not due to heat exposure. For instance, socioeconomic status is known to affect birth outcomes, but is also likely to affect both place of residence and timing of conception [64]. A link with place of residence may be particularly important in the NT, given that most non-Aboriginal people live in the capital city of Darwin, whereas most Aboriginal people live in remote communities [65], many of which face higher levels of heat exposure than Darwin. Similarly, we know that there are seasonal risks in addition to heat exposure, such as disease prevalence and economic conditions, which contribute to month-to-month variations in birth outcomes [37].

Our analytical approach is to estimate the causal effect of heat exposure by isolating fluctuations in heat that are exogenous to these (possibly endogenous) sources of time- and place-based variation. Exposure to such exogenous variation is beyond individuals' control, and cannot be anticipated more than a few days in advance. This variation in exposure is, therefore, as good as random.

To do this, we specify a regression model with fixed effects for time and place. This approach is discussed in detail in Dell et al.'s review [40], and has been widely used in prior research.

As is common practice, we allow for the effects of heat exposure to vary by trimester; this allows for the possibility, as discussed in the Conceptual framework, that the effects of heat exposure may depend on its timing – for instance, exposure in the first trimester may increase the risk of pre-eclampsia [27], and exposure in the third trimester may bring forward labour [33].

We therefore estimate the following linear fixed effects model:

$$preterm_i = b_0 + \sum_{k}^{3} b_1^k heat_{tj}^k + b_2 X_i + \gamma_{my} + \theta_{jmi} + \in_{ij}$$

(1)

where $preterm_i$ is an indicator of whether baby $i$, who was conceived on date $t$ (in month $m$, year $y$), was born preterm or not. $k$ is an index of the three trimesters of pregnancy. $X$ is a set of individual-level covariates, which includes an indicator for whether it is the mother's first pregnancy, the age of the mother (in 5-year age bands), and an indicator for whether the baby is Aboriginal. Standard errors are clustered at the location level. We identify location, $j$, based on the mother's suburb/community of residence, which we have geocoded, and grouped into locations of maximum 50-mile distance between points using cluster analysis. There are 106 such clusters in our data, with an average of 311 births per cluster.

*Heat* represents any one of the five heat metrics outlined above. They are based on daily weather observations from small-area location $j$, which are summarised over each trimester based on exact date of conception $t$. Our coefficients of interest are $b_1^k$.

In specifying the fixed effects, we follow prior studies in using fixed effects for the month-year of conception, $\gamma_{my}$, and for the mother's place of residence, interacted with the month of conception and the baby's sex, $\theta_{jmi}$. Interacting location fixed effects with month and sex allows for the possibility that there may be different seasonal patterns in birth outcome across locations – this may be the case in the NT given, for example, that some communities experience regular flooding in the wet season, and hence their experience of the wet season may be different from other communities that do not flood. Furthermore, we know that different types of exposures may affect male and female fetuses differently [66]. Interacting these location- and season-specific effects with sex allows for this possibility.

To illustrate what we are comparing with these community-month-sex fixed effects, consider two boys born in Darwin in January of the same year. Both boys were conceived in the same location (Darwin) and the same month (April), so

they share the same location-month-sex fixed effect, though they were conceived two weeks apart. They also share the same month-year fixed effects. However, their mothers experienced different heat exposure during pregnancy due to particularly warm weather occurring in January while one baby remains in utero but the other has already been born. The fixed effects allow us to compare outcome for these two boys, attributing any difference to the variation in heat exposure their mothers experienced, while holding constant all other factors that vary systematically by location, month of conception, and baby's sex.

This day-to-day variation in weather within the same location-month-sex group is the source of identification in our model. It represents truly exogenous variation—mothers cannot anticipate or control whether their pregnancy will coincide with an unseasonably hot days versus a cooler days within the same general season and location.

Our choice of interacted fixed effects follows prior studies [4,67]. That said, S1 Table shows that our estimates are similar regardless of the specifics of how we define these fixed effects.

**Machine learning considerations.** In this analysis, we set out to select a preferred heat metric from among many potential regressors. To do this, we initially considered use of machine learning – and in particular lasso regression. However, in this case we determined that a machine learning approach was not appropriate because of the low explanatory power of heat exposure variables relative to other factors that determine the risk of preterm birth. As we show in our analysis and address in our discussion, heat exposure explains a very small share of variation in outcomes. The differences in explanatory power between alternative metrics are smaller still (see Discussion section below). Hence, when we apply a lasso regression, the penalty structure prioritises parsimony over detecting small but meaningful effects – and eliminates heat exposure variables. We are aware that double machine learning approaches can help to address the regularisation bias. But we determined that, given the goal of our analysis was to make general recommendations for how heat exposure should be measured, we favoured a more direct approach to comparing alternative metrics.

### 3.6. Our approach to selecting a preferred heat metric

Part of our goal in this analysis is descriptive: to run the same model sequentially with each heat metric, and learn how choice of heat metric affects our conclusions around the impacts of heat exposure on preterm birth. To do this, we present regression coefficients for the same model, using each of our five heat metrics.

However, we also wish to select a preferred heat metric. Out of our five candidate metrics, we want to know which one best captures the impact of heat exposure on preterm birth in the NT. In doing so, we face a challenge: because each regression uses different heat exposure metrics, the magnitudes and statistical significance of the resulting coefficients cannot be directly compared with each other.

We therefore take a two-stage approach to selecting a preferred metric. First, we present an F-test of joint significance of the regressors included in each heat exposure metric. If the regressors are not jointly statistically significant, we exclude the metric from further comparison. Second, we compare goodness of fit, for the remaining metrics.

We use two measures of goodness of fit. First, the R-squared. This tells us how much variation in the outcome is explained by the regressors. A standard R-squared increases when additional regressors are added. To account for different numbers of regressors in each metric, we use an adjusted R-squared: this penalises additional regressors. A higher adjusted R-squared, therefore, indicates that the model fits the data better, when comparing among models with different numbers of regressors. Second, we present the Akaike Information Criterion (AIC). The AIC helps us to compare several alternative, non-nested models, on a single measure: it is calculated as the likelihood of the model (estimated using maximum likelihood), penalised for the number of regressors. It therefore weighs up model fit with model complexity, allowing us to compare multiple working hypotheses against each other [68]. In their review on variable selection, Heinze, Wallisch and Dunkler [69] recommend using AIC in cases such as ours, where theory supports a relatively small set of competing models, and we want to select between them. In comparing AIC scores, a lower score indicates better fit.

**Table 2. Measures of model fit for each heat metric.**

|  | F test for joint significance of heat exposure terms | Adj R² | AIC |
|---|---|---|---|
| Benchmark | 5.36 | 0.036 | 13214 |
| Trimester average | 2.23 | 0.035 | 13251 |
| Heatwave count | 1.77 | 0.035 | 13267 |
| Max and min | 11.49 | 0.037 | 13174 |
| Wet bulb | 8.95 | 0.035 | 13233 |

Each row of this table represents results from a regression using the indicated heat metric. The first column in this table shows F-statistics for an F-test of joint statistical significance of all terms in the heat exposure metric. The second column, Adj $R^2$, represents the adjusted R-squared, for which a higher value represents better fit. Note that here adjusted R-squared is 'within' the community × sex × month fixed effects (which are absorbed using 'areg' in Stata). The third column, AIC, is the Akaike Information Criteria, for which a lower value indicates better fit. Sample size: 34,258.

## 4. Results

### 4.1. Goodness of fit comparison

Table 2 presents a comparison of how well each heat metric fits the data.

Across our five metrics, we can first exclude 'Trimester average' and 'Heatwave count' from comparison: in an F-test of joint significance at the 5% level of significance, we would fail to reject the hypothesis that the coefficients in these metrics are jointly equal to zero, hence that they do not help to explain variation in preterm birth rates.

We then have three remaining metrics. Across all metrics, the adjusted R-squared is low and varies little, ranging between 0.035 to 0.037. However, the metrics produce very different AIC scores. On both R-squared and AIC measures, the 'Max and min' metric performs better: it has a slightly higher adjusted R-squared and substantially lower AIC.

We separately consider whether a combination of metrics provides a better fit – for instance, combining the 'Max and min' metric, the 'Wet bulb' metric and 'Heatwave count' metric together. However, while some of the additional coefficients are statistically significant, they do not meaningfully improve model fit (S2 Table), and interpretation of individual coefficients becomes very difficult.

We therefore select the 'Max and min' metric as our preferred heat exposure metric.

### 4.2. Coefficient estimates

Having identified the 'Max and min' metric as our preferred one, we now turn to a presentation of the coefficients for the regressors in each heat metric. This tells us both the estimates from our preferred model, and the potential consequences, in terms of inferences we may draw, from selecting a non-preferred metric.

We present estimates graphically, showing coefficients and their 95% confidence intervals. Tables with all coefficients cited and their standard errors can be found in S3–S10 Tables.

**Benchmark heat metric.** Fig 3 presents estimates from our benchmark metric. We can see little impact from heat exposure in the first trimester. However, in the second and third trimesters we find that exposure to cooler temperatures (below our omitted category of 25–30 degrees) contributes to lower risk of preterm birth; we see no additional impact of heat exposure above 30 degrees. This suggests that it may be predominantly heat strain (i.e., prolonged exposure to moderate heat), instead of heat stress (i.e., exposure to extreme heat) affecting pregnancies in the second and third trimesters.

If it is the case that maternal heat strain affects the baby through reduced flow of nutrients to the placenta [24], this could explain why we see positive effects of cooler temperatures in the second and third trimesters, but not the first trimester: in the first trimester, the fetus receives nourishment from the yolk sack and not the placenta, and is therefore less

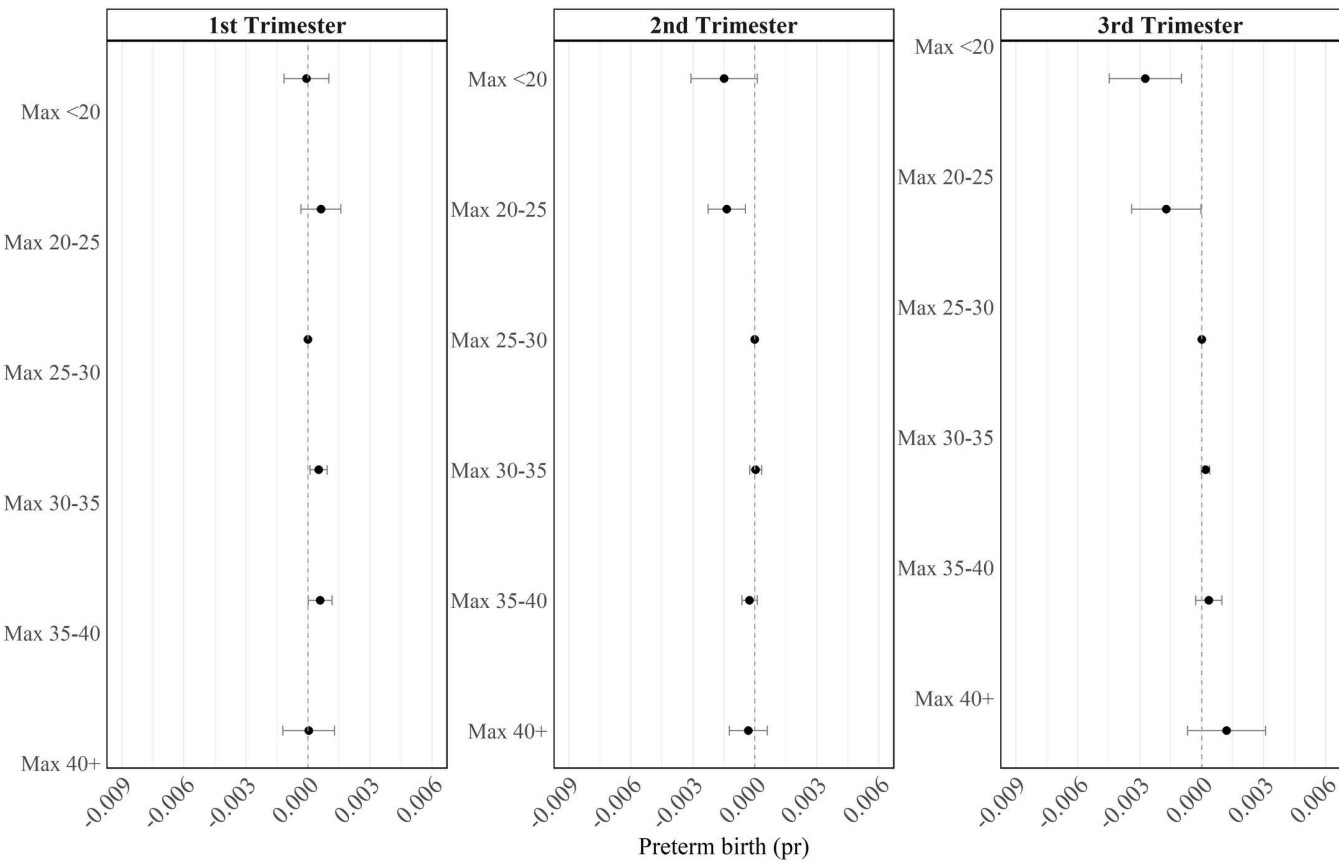

**Fig 3. Benchmark heat metric.** This figure shows the regression coefficients and 95% confidence intervals from Equation 1, where the heat metric is a count of the number of days within each trimester that the daily maximum temperature fell into the following ranges: < 20, 20–25, 25–30, 30–35, 35–40, 40 + . The omitted category is 25–30 degrees. Sample size: 34,258.

reliant on maternal blood flow for nutrients. When the placenta takes over around the beginning of the 2nd trimester, this is when we might expect an impact of heat strain.

**Average temperatures.** Fig 4 shows estimates from the 'average temperature' metric. Note the difference in scale from Fig 3 due to differences in units: a 1 degree increase in average temperatures represents more heat exposure than a single day of extreme heat. For example, in a trimester with average temperatures of 30 degrees, a 1 degree increase in the average would represent over one week of temperatures of 40 or above.

Fig 4 shows that a 1 degree increase in average temperatures in the third trimester leads to a 0.007 percentage point increase in the probability of preterm birth. Although we saw some statistically significant impacts of heat exposure in the first and second trimesters of the benchmark model, these are not evident here.

**Heatwave count.** Fig 5 shows estimates from our 'heatwave count' metric. We find that while the coefficients are positive (indicating that experiencing an additional heatwave in utero increases the risk of preterm birth), they are not statistically significant. Though this is not the case for all outcomes – we find statistically significant coefficients on average birthweight, with each additional heatwave in the second trimester associated to a 5.7 gram reduction (S5 Table).

The fact that we do not find statistically significant effects on preterm birth makes sense given that, in our benchmark model, much of the effect of heat exposure came through improved outcomes from exposure to cooler temperatures (below 25–30 degrees), and not from extreme heat conditions that would contribute to a heatwave.

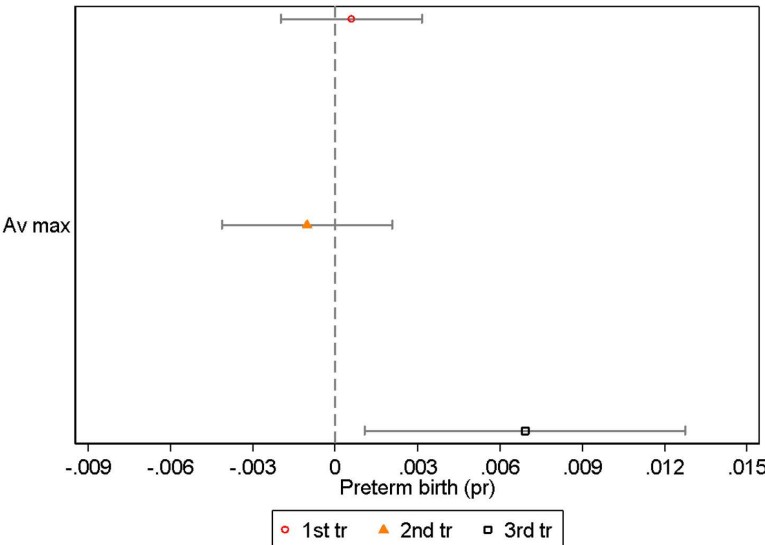

**Fig 4. Trimester average.** This figure shows the regression coefficients and 95% confidence intervals from Equation 1, where the heat metric the trimester-average of daily maximum temperatures. Sample size: 34,258.

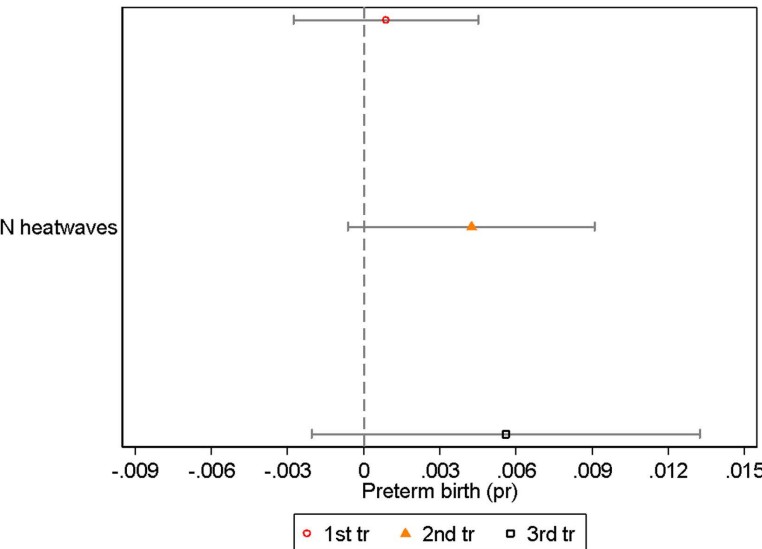

**Fig 5. Heatwaves.** This figure shows the regression coefficients and 95% confidence intervals from Equation 1, where the heat metric is a count of the number of heatwaves. Heatwaves are defined as 3 or more consecutive days of unusually hot weather relative to usual conditions within a location – using on the Australian Bureau of Meteorology's method. Sample size: 34,258.

**Max and min (preferred metric).** The 'Max and min' metric – our preferred metric, based on goodness of fit – adds minimum temperature exposure bands to our benchmark metric. We see similar patterns to the benchmark metric, though the addition of minimum temperatures reveals further effects of prolonged heat exposure – i.e., increased risk of preterm birth for babies exposed to more days when minimum temperatures do not fall below 20 degrees (Fig 6).

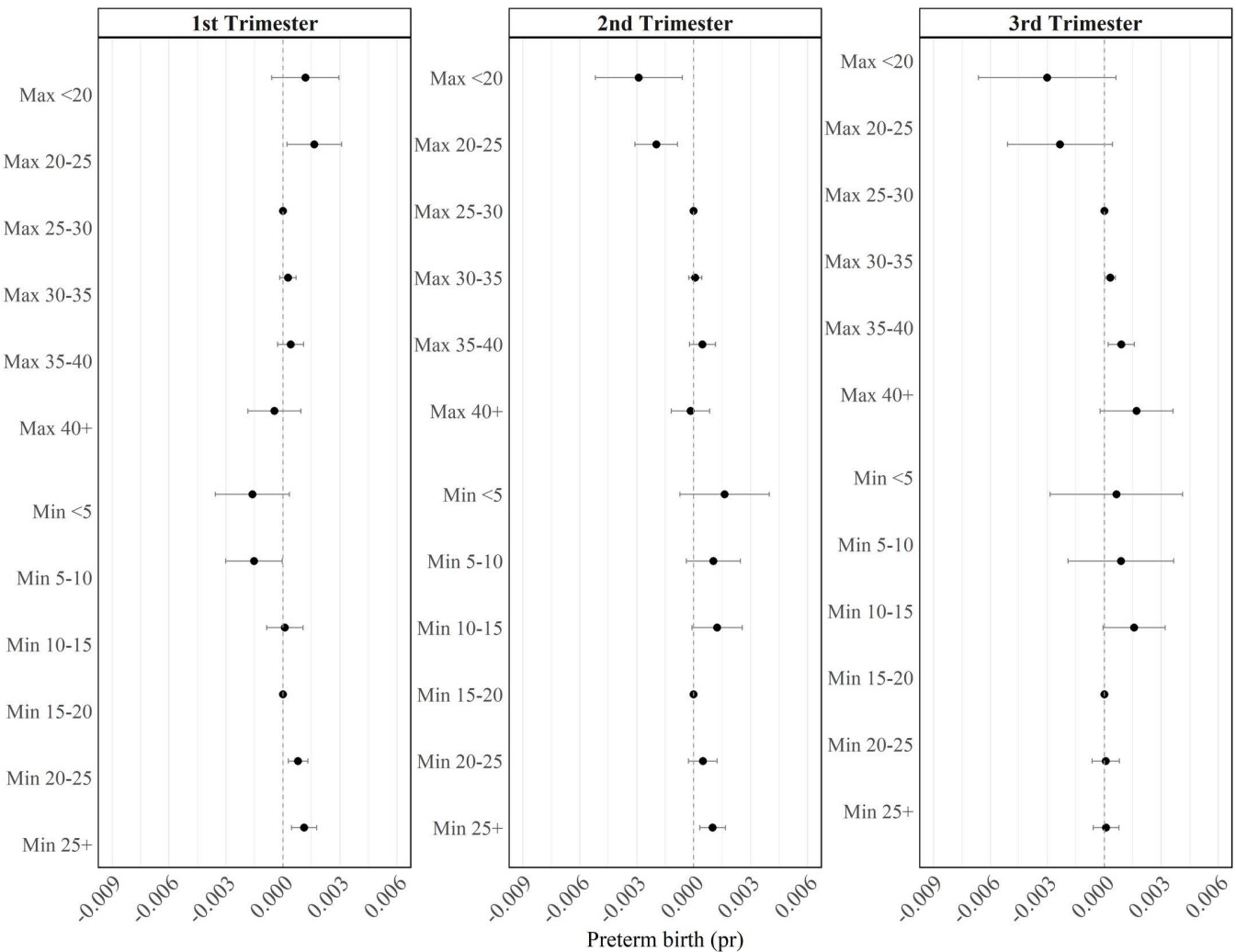

**Fig 6. Max+min.** This figure shows the regression coefficients and 95% confidence intervals from Equation 1, where the heat metric is a count of the number of days within each trimester that the daily maximum temperature fell into the following ranges: <20, 20–25, 25–30, 30–35, 35–40, 40+, and a count of the number of days within each trimester that daily minimum temperatures fell into the following ranges: <5, 5–10, 10–15, 15–20, 20–25, 25+. The omitted categories are 25–30 degrees (max) and 15–20 (min). Sample size: 34,258.

In the first trimester, high minimum temperatures of 20 or above increase the risk of preterm birth, while low minimums (below 10 degrees) reduce the risk; we no longer see a statistically significant impact of high maximum temperatures.

In the second trimester, we continue to see a protective impact of low maximum temperatures (below 25 degrees), and these estimates are larger in magnitude here than in the benchmark metric. We also find that high minimum temperatures (25 degrees or above) increase the risk of preterm birth.

In the third trimester, we now see that high maximum temperatures (between 30–39.99 degrees) increase the risk of preterm birth, though with no statistically significant impact of minimum temperatures. Some of these changes in estimates may be due to multicollinearity, given that low values of maximum and minimum temperatures each have similar coefficients in the third trimester when modelled separately (S3 Table).

As this is our preferred metric, we ran our analysis separately for Aboriginal and non-Aboriginal babies, to assess whether the effects of heat exposure are different amongst these different populations (see S7 Table). We find estimates

are similar in both groups, though among Aboriginal children, the only statistically significant estimates are for the third trimester: exposure to cooler maximum temperatures reduces the risk of preterm birth, and exposure to temperatures of 35–40 degrees increases the risk. In contrast, the effects of higher maximum and minimum temperatures in the first and second trimesters are statistically significant among non-Aboriginal children, but there is little effect of third trimester exposure. It may be that Aboriginal communities and cultural practices are better adapted to hot weather, leading to little impact of a marginal increase in heat exposure. It may also be the lack of statistically significant effects reflects the greater diversity among Aboriginal mothers in their exposures to heat due to the varied locations of Aboriginal communities – leading to estimates that are generally of similar magnitude to those for non-Aboriginal mothers, but less precisely estimated [37].

**Wet bulb.** Fig 7 shows coefficients for analysis using our 'Wet bulb' metric – which is analogous to the 'Max and min' metric, but constructed using wet bulb temperatures instead of air temperature. We see little statistically significant

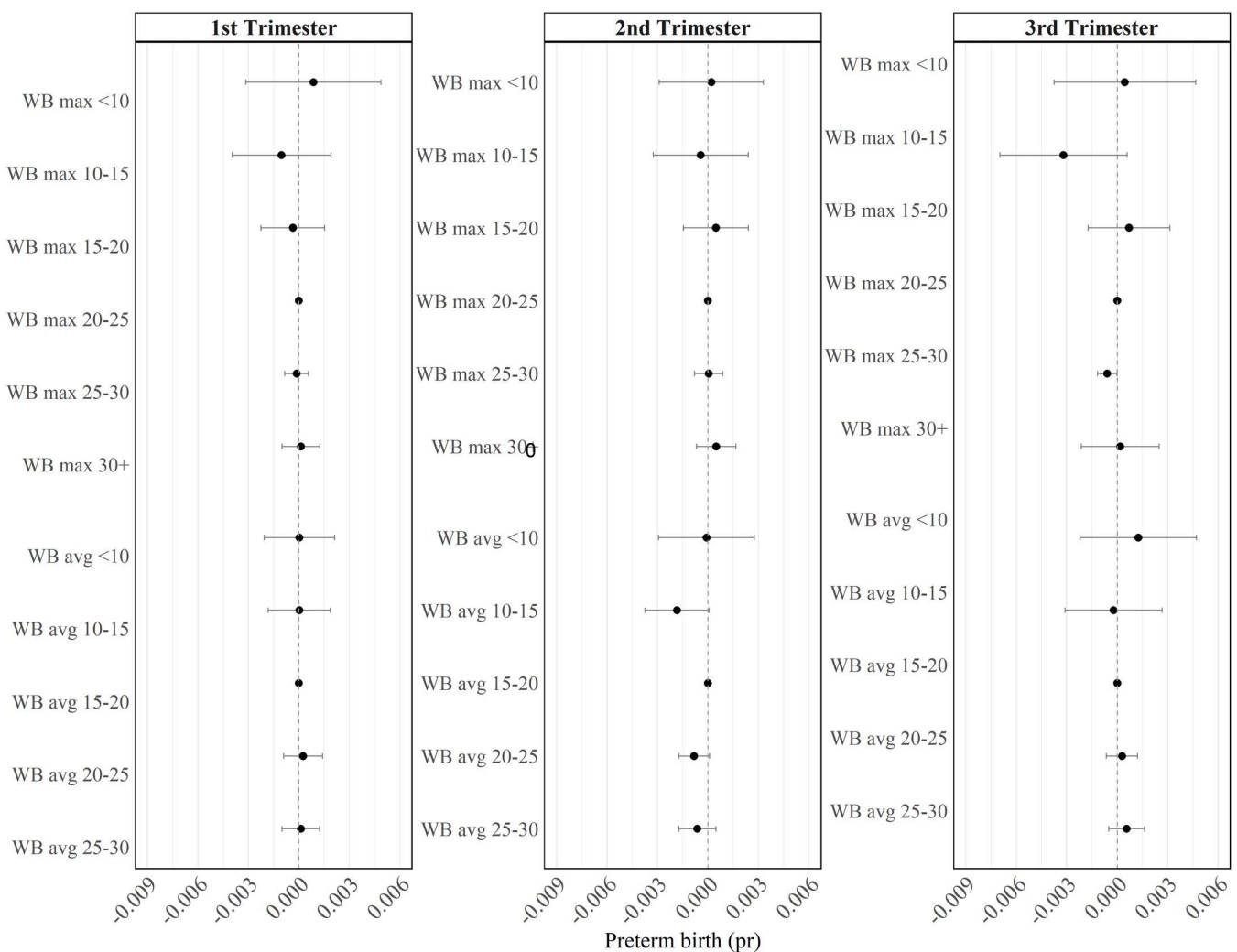

**Fig 7. Wet bulb.** This figure shows the regression coefficients and 95% confidence intervals from Equation 1, where the heat metric is a count of the number of days within each trimester that the daily maximum wet bulb temperature fell into the following ranges: < 10, 10–15, 15–20, 20–25, 25–30, 30 +, and a count of the number of days within each trimester that daily average wet bulb temperature fell into the following ranges: < 10, 10–15, 15–20, 20–25, 25 +. The omitted categories are 20–25 (max) and 15–20 (avg). Sample size: 34,258.

relationship between wet bulb temperatures and preterm birth: the only statistically significant coefficient suggests that high wet bulb temperatures (25 to 29.99 degrees) in the third trimester decrease the risk of preterm birth – that is, an effect in the opposite direction to that suggested by all other metrics.

Fig 7 presents both average and maximum daily wet bulb temperatures in a single metric. But sensitivity testing shows that our conclusions would be similar if we were to construct our metric using maximum or average wet bulb values separately. As shown in S9–S10 Tables, we find small statistically significant coefficients on low values of daily maximum wet bulb temperatures in trimesters 1 and 3, but this model has low predictive power.

### 4.3 Do these findings differ by climate zone?

As described in the Introduction, the relationship between various measures of heat exposure – and hence the extent to which selection of heat metric matters – varies across climate zones. We therefore repeat our analysis, but running models separately for the tropical climate zone in the north of the NT, and the arid climate zone in central NT.

We see some differences in coefficients between the climate zones (S11 Table). For instance, while high minimum temperatures in both the first and second trimesters significantly increase the risk of preterm birth in the tropical climate zone, we do not find this in the arid climate zones. In addition, while high maximum temperatures increase the risk of preterm birth in both climate zones, the magnitudes of these coefficients are at least twice as large in arid climate zones.

We may have expected the metric based on wet bulb measures to be a better fit in the tropical climate zone. However, we do not find this (S12 Table). Across both climate zones, the 'Max and min' metric provides the best fit (S13 Table).

## 5. Discussion

At a high level, it is clear from Figs 3–7 that choice of heat metric can have a substantial impact on our conclusions around how much prenatal heat exposure affects the risk of preterm birth. Our preferred metric,'max and min', shows that exposure to both extreme heat (e.g., additional days with maximum temperatures of 35–39.99 degrees) and prolonged exposure to more moderate heat (e.g., additional days with minimum temperatures of 20 degrees or higher) both increase the risk of preterm birth. This general pattern is evident in both our preferred 'max and min' metric, and in the benchmark metric, which is more commonly used in prior research. However, our preferred metric fits the data slightly better. This finding is consistent with other research showing daily minimum temperatures affect health independently of maximum temperatures [70].

However, perhaps more importantly, we find that in the NT, and likely in other contexts as well, the choice of heat metric can determine whether we conclude that prenatal heat exposure affects birth outcomes at all. In particular, we find no statistically significant impact of the number of heatwaves during pregnancy on the risk of preterm birth. In addition, while our preferred metric finds some impact of higher temperatures exposure in all trimesters, these effects are not linear, and hence are not evident in the first and second trimesters in our 'Trimester average' metric, which imposes linearity.

Somewhat surprisingly, we also find that our 'Wet bulb' metric, although it allows for the nonlinearities we find in our preferred metric, does not capture any clear effect of heat exposure. We discuss possible reasons for this below in Section 5.2.

Having established a preferred heat metric, we now turn to a discussion of the practical implications of using non-preferred metrics, and our recommendations for metric selection in future research.

### 5.1. Practical implications of using non-preferred heat metrics

The analysis we have presented above demonstrates the statistical significance and model fit of five alternative metrics. But the varied scales of the coefficients make it difficult to compare the practical significance of these different estimates. While we do find one metric that outperforms the alternatives on model fit, the differences appear marginal – our preferred

'max and min' metric has an adjusted R-squared that is just 0.001 units higher than the benchmark metric. Furthermore, the explanatory power of even our preferred heat metric is low – clearly there are many more proximate factors, not included in our models, that explain the vast majority of variation in preterm birth rates. That said, while the effects of pre-natal heat exposure are small for any one individual, these impacts are important at the population level. This is particularly so in the context of high and rising levels of heat exposure.

To quantify the implications of metric choice, we use our regression estimates to generate predictions of the impact of typical within-year differences in heat exposure on preterm birth rates.

We generate two sets of predictions: first, the hypothetical case where all babies in our analysis data were exposed to heat typical of the hottest 9 months of the year in utero (born in May), and second, the hypothetical case where all babies were exposed to heat typical of the coolest 9 months of the year (born in November). We then compare the difference between these two predictions in Fig 8. Our methodology is outlined in S2 Appendix.

In our preferred 'Max and min' heat metric, we estimate that if all babies were in utero during the coolest 9 months of the year, preterm birth rates would be 9 percent, and if all babies were in utero during the hottest 9 months of the year, preterm birth rates would be 13.5 percent. This is a 4.5 percentage point difference.

In contrast, the alternative metrics imply effects of less than half this size: we estimate a 2 percentage point difference using the benchmark metric, and smaller differences from the other metrics. Therefore, choosing a non-preferred metric means not only that we may mis-understand the relationship between heat exposure and preterm birth, but also that we may severely under-estimate heat's implications for population health.

Of course, there is uncertainty around these predictions: the difference between these two predictions with our preferred metric is not statistically significant at the 95% level. However, based on the large body of research and our own analysis, we know that it is more likely than not that hot weather causes a substantial worsening in health at birth, and a 4.5 percentage point effect is our best estimate of the impact of normal seasonal variation in the NT. To put this in context, an effect of this magnitude is equivalent to intensive rates of smoking during pregnancy [18].

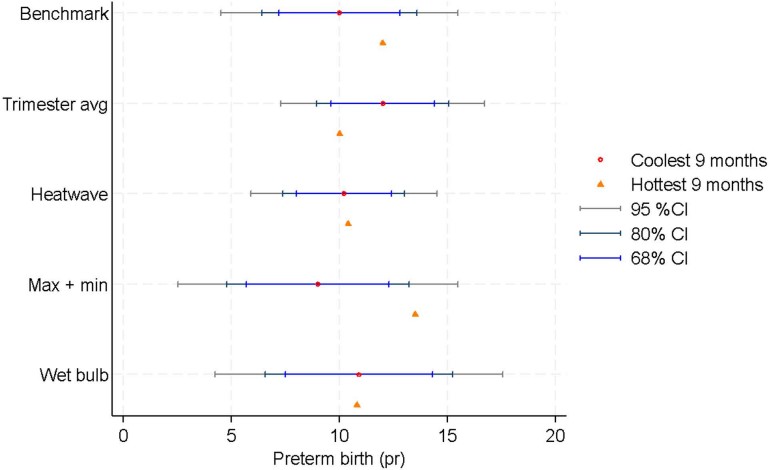

**Fig 8. Model predictions with 95%, 80% and 68% confidence intervals.** Predictions shown for all heat metrics, and represent the models' predicted rate of preterm birth for a population exposed to typical conditions in the coolest 9 months of the year in utero, compared with a population exposed typical conditions in the hottest 9 months of the year in utero. The confidence intervals are shown around the estimate for the coolest 9 months for each metric.

## 5.2. Recommendations for future analysis

In our analysis, we have compared five different approaches to defining and measuring heat exposure – we call these 'heat metrics'. We have found that a metric based on air temperature which allows for non-linear effects of increased temperatures, and includes both daily minimum and maximum temperatures, provides the best fit. Our findings lead us to two general recommendations for future research.

First, it is very unlikely that the true relationship between heat exposure and birth outcomes is the one implied by either the 'Trimester average' or the 'Heatwave count' metric. As outlined in our Conceptual framework, we have good reason to believe that both moderate and extreme heat during pregnancy can affect birth outcomes, possibly in different ways. Therefore, at a minimum, our models should allow for these nonlinear effects. While some studies have found that the 'Trimester average' and 'Heatwave count' metrics nevertheless produce statistically significant estimates, we have found that they will not necessarily do so. Furthermore, as shown in Fig 8, even among alternative metrics where the regressors are statistically significant, the implied impact of heat exposure from a more limited metric may differ from a more flexible metric by orders of magnitude. Therefore, where researchers have the choice, we recommend against using the trimester average or heatwave count metrics.

Second, in the population we analyse, we find that metrics based on air temperature, and not on wet bulb temperatures, better capture the effects of heat exposure – even though most of this population lives in a humid, tropical climate zone. While surprising, our findings are not unique; many epidemiological studies have also found little impact of humidity on health outcomes [16]. In understanding this finding, we cannot rule out issues with data quality and choice of measure. Our wet bulb data are derived from satellite images, and not directly measured at a weather station. This is the only metric incorporating humidity that is available within our region – weather station data do not cover the whole of the NT, meaning we do not have the underlying data series needed to construct alternative heat indices. It may be that, if they were available, alternative heat indices that combine heat and humidity could have more explanatory power than wet bulb measures. That said, we note that studies using reanalysis wet bulb data for other outcomes, such as LoPalo's [71] analysis of the impacts of heat on worker productivity, do find wet bulb measures to be more predictive than ambient temperatures.

However, our findings may also reflect real, complex relationships between heat, humidity and health in a tropical climate. The role of climate may explain, for instance, why a recent study from Western Australia finds clear effects from a metric based on a heat index combining temperature and humidity [72]. While some parts of Western Australia have a tropical climate, the vast majority of the population in that study lives in Perth, which has a dry, Mediterranean climate, and hence humidity will play little role in determining variation in the index.

It may be that the interplay of heat and humidity is not well captured by a single data series, whether it is wet bulb or another heat index. In Darwin, for instance, days with higher humidity and rainfall typically do not reach particularly high maximum temperatures; in fact, humidity and air temperatures are negatively correlated with each other (see S14 Table and S3 Fig). But because humidity is a contributor to wet bulb measures, wet bulb temperatures are mechanically higher on cooler, rainy days. Therefore, it is possible that hot, humid days with higher wet bulb temperatures can, in theory, contribute to greater heat strain. But the same wet bulb temperatures can also be the result of very humid, cooler-than-average and rainy days, which provide some relief from high temperatures. This may explain why our estimates suggest that higher wet bulb temperatures in the third trimester can actually reduce preterm birth rates. It may also be that the same level of humidity feels different on a day with intense rainfall (i.e., the wet season) compared with days with no rainfall (i.e., the build up to the wet season). Future research could analyse air temperature, humidity and rainfall, and the interaction between the three as separate regressors, to test whether this yields more informative estimates.

Our takeaway is that in the NT, as well as in many other contexts where the only data available are satellite or reanalysis data, heat exposure metrics based on air temperature may well provide more explanatory power than metrics based on wet bulb temperatures. However, future research could compare a range of heat indices in a location where data are known to be of high quality and where humidity and extreme heat co-occur: this would help to more definitively assess

whether the lack of consistent effects of humidity that we and others find is due to data quality, or to a mis-specification of the relationship between heat, humidity and health.

## 6. Conclusions

We have found that in the Northern Territory of Australia, the way that we define and measure heat matters a lot for our conclusions on how prenatal exposure affects health at birth. Choice of metric can mean the difference between finding no effect, finding a large, statistically and practically significant effect or finding a statistically significant effect, but which is practically insignificant given normal fluctuations in heat exposure.

In our context, we find that a heat metric allowing for non-linear effects of both minimum and maximum daily air temperature provides the best fit. Such a metric may be particularly valuable in tropical climates, where correlations between daily maximum and minimum temperatures are low (Fig 1).

Recent methodological discussions by Brimicombe et al. [15] and Leung [17] have recommended constructing heat metrics based on both air temperature and humidity, moving away from use of air temperature alone. In contrast, we find that such a metric provides a worse fit, and as a result, under-estimates the impact of typical variation in heat exposure. We do not know whether this is because of data quality or the underlying relationship between heat, humidity and health outcomes. However, the implication is that, at least with data that are currently widely available, researchers should be cautious when applying such metrics. Furthermore, in climates where these measures are highly correlated with air temperature, meaning that choice of underlying data series matters less, researchers' decisions around how to allow for non-linear and within-day temperature variation may be more important than their choice of underlying data series.

Overall, our estimates show that both moderate heat and more extreme heat contribute to poorer health at birth. The relative contributions of different heat levels and the relevant timing of exposure are not stable across different heat metrics or climate zones, and therefore we do not draw any strong conclusions on what levels and timing of heat exposure matter most. A range of unobserved factors are likely to influence these thresholds in any given context. For instance, medical advice, culture, work norms for pregnant women, and adoption of air conditioning and other adaptations will affect the size of this relationship and the relative contributions of heat stress versus heat strain-related conditions. Many of these factors are likely to vary on important dimensions that we cannot measure – for instance, across socioeconomic status and income levels. Consequently, while our analysis is informative, it should be replicated in diverse contexts to test the generalisability of our findings. To understand these mechanisms further, it would also be helpful to learn more about how pregnant people perceive and respond to heat exposure risks. For instance, Kc et al. [73] conduct a survey assessing perceived risks of heat exposure among pregnant people in Nepal – the same survey could be carried out in other contexts to better understand behavioural responses to heat exposure.

While uncertain, our central estimate is that normal levels of heat experienced in the hottest months of the year could be contributing to a 4.5 percentage point higher risk of preterm birth in the NT – rates 45% higher than we would predict in babies in utero during the coolest 9 months of the year. The scale of these fluctuations, and their longer-term implications for children's development, may be even greater in lower-resource settings with similar climates, where healthcare and remedial neonatal care are less readily available. Furthermore, as climate change leads to additional exposure for populations previously unexposed to extreme heat or prolonged periods of moderate heat, pregnant people who do not have the habits and adaptations that are already present in regions such as the NT may experience more acute effects.

We identify two important directions for further research. First, analysing the long-term health impacts of heat exposure, looking beyond measures of health at birth – which are themselves imperfect indicators of fetal health and development [74]. As noted in our conceptual framework, outcomes like preterm birth are informative, but not all children who are born preterm will experience poorer health later in life, and similarly, not all children who experience poorer health later in life as a result of in utero heat exposure will have been born preterm. Second, analysing dose-response relationships between

heat exposure and birth outcomes, and investigating whether there may be cumulative effects of continued exposure and/or periods during pregnancy where heat exposure poses the greatest risk.

## Supporting information

**S1 Appendix. NASA satellite vs observational weather data.**
(DOCX)

**S2 Appendix. Prediction methodology.**
(DOCX)

**S1 Fig. Scatter plot of daily average air temperatures and daily average wet bulb temperatures, 2020–2023.**
(DOCX)

**S2 Fig. Scatter plot of daily maximum and minimum temperature in Alice Springs, 2020–2023.**
(DOCX)

**S3 Fig. Scatterplot of daily rainfall with average wet bulb temperature and maximum temperature in Darwin.**
(DOCX)

**S1 Table. Regression estimates with alternative specifications of fixed effects.**
(DOCX)

**S2 Table. Model build with multiple metrics together.**
(DOCX)

**S3 Table. Regression coefficients for benchmark metric and additional outcomes.**
(DOCX)

**S4 Table. Regression coefficients for average max temperature metric with additional outcomes.**
(DOCX)

**S5 Table. Regression coefficients for heatwave metric with additional outcomes.**
(DOCX)

**S6 Table. Regression coefficients for max and min metric with additional outcomes.**
(DOCX)

**S7 Table. Regression estimates for probability of preterm birth by Aboriginal status, preferred metric.**
(DOCX)

**S8 Table. Regression coefficients for wet bulb metric with additional outcomes.**
(DOCX)

**S9 Table. Average wet bulb piecewise linear with additional outcomes.**
(DOCX)

**S10 Table. Maximum wet bulb (WB) piecewise specification with additional outcomes.**
(DOCX)

**S11 Table. Regression coefficient estimates by climate zone – air temperature.**
(DOCX)

**S12 Table. Regression coefficient estimates by climate -wet bulb metric .**
(DOCX)

**S13 Table. Measures of model fit for each heat metric – tropical climate zone.**
(DOCX)

**S14 Table. Correlations coefficients among various weather metrics in Darwin.**
(DOCX)

## Acknowledgments

The research data used in this study was provided by the Child and Youth Development Research Partnership (CYDRP) between the Menzies School of Health Research and the Northern Territory Government. We thank the South Australia-Northern Territory Datalink data integrating authority for facilitating linkage of the multiple datasets made available through CYDRP. We also thank Steve Guthridge, Stephen Jenkins, Berkay Özcan and Dilhan Perera for their feedback on this manuscript.

## Author contributions

**Conceptualization:** Mary-Alice Doyle.

**Data curation:** Bernard Leckning.

**Formal analysis:** Mary-Alice Doyle.

**Investigation:** Mary-Alice Doyle.

**Methodology:** Mary-Alice Doyle.

**Project administration:** Bernard Leckning.

**Writing – original draft:** Mary-Alice Doyle.

**Writing – review & editing:** Mary-Alice Doyle, Bernard Leckning.

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
