## [Decision Letter · Decision Letter 0]

22 May 2025

PONE-D-25-13859The relationship between prenatal heat exposure and birth outcomes: how much does the heat metric matter?PLOS ONE

Dear Ms. Doyle,

Thank you for submitting your manuscript to PLOS ONE. I have now received referee reports from two economists who work on the health impacts of climate change. Based on their recommendations and my own reading of the manuscript, I have decided to offer you a chance to revise the manuscript to meet the criteria for eventual publication.  Please submit your revised manuscript by Jul 06 2025 11:59PM. If you will need more time than this to complete your revisions, please reply to this message or contact the journal office at plosone@plos.org . Please include the following items when submitting your revised manuscript:

We look forward to receiving your revised manuscript.

Kind regards,

Xinde James Ji

Academic Editor

PLOS ONE

Journal Requirements:

“London School of Economics PhD Studentship”

Additional Editor Comments:

Both reviewers appreciate the importance of your research questions and find merits in your general approach. In addition to the general comments, Reviewer #1 suggested alternative, machine-learning-based approaches to the variable selection problem (e.g. lasso), which is standard in the recent econometrics/statistics literature and can be reconciled with a causal interpretation (see work by Susan Athey, Victor Chernozhukov, et al.) I agree with the reviewer and think there are merits in trying that.

Reviewer #1 also suggested a band-based approach standard in the climate impact literature, which can be combined with a wet-bulb-based temperature metric (see e.g., Melissa Lapalo, 2023 AEJ: Applied). I'll leave that to you as to whether you want to implement such an approach. 

While revising your manuscript, please consider all issues mentioned in the reviewers' comments carefully: please outline every change made in response to their comments and provide suitable rebuttals for any comments not addressed. Please note that I intend to go back to the referees for additional feedback.

Reviewers' comments:

Reviewer's Responses to Questions

**Comments to the Author**

1. Is the manuscript technically sound, and do the data support the conclusions?

Reviewer #1: Partly

Reviewer #2: Partly

2. Has the statistical analysis been performed appropriately and rigorously? 

Reviewer #1: Yes

Reviewer #2: Yes

3. Have the authors made all data underlying the findings in their manuscript fully available?

Reviewer #1: Yes

Reviewer #2: Yes

4. Is the manuscript presented in an intelligible fashion and written in standard English?

Reviewer #1: Yes

Reviewer #2: Yes

5. Review Comments to the Author

Reviewer #1: - Manuscript Number: PONE-D-25-13859

- Title: The relationship between prenatal heat exposure and birth outcomes: how much does

the heat metric matter?

Overview and general recommendation

This paper studies the impact of heat exposure on preterm birth using administrative data from the Northern Territory of Australia. It highlights the importance of heat metrics on the estimation results and presents an exercise in the metrics selection procedure. I find this paper's findings potentially important, and its perspective on the measurement issue is innovative. The paper is also well written and has the potential to make a significant contribution to the environmental econometrics literature.

I have some concerns regarding the methodology and research design for the editor to consider. I elaborate on my comments and suggestions below:

Major comments

1.Criteria for selecting preferred model selection.

The main potential contribution of this paper is to provide a model selection approach among the commonly used heat exposure metrics. Table 2 presents the most important results. However, I am somewhat reluctant to agree with the three selection criteria that the authors use. They employ a two-step selection method: first, they exclude the inferior candidates using Joint F-statistics; then, they select the best one based on Adjusted R² and AIC. I have three comments.

Econometrically Distinguishable? The first issue is that the adjusted R² and AIC for alternative metrics are too close to strongly support which one is better. For the adjusted R², all five metrics fall between 0.035 and 0.037. I would expect a much larger difference in R², given that Figure 1 shows that in your studied context, the daily maximum and minimum temperatures are not correlative. I would anticipate that the five candidates could explain a significant amount of variation.

Additionally, this absolute magnitude is surprisingly small, especially considering that the authors have controlled for rich fixed effects. This may indicate that other factors are interpreting the majority of the variation in the data. More importantly, if you include some of these variables, would the current ranking persist?

Apple-to-Apple Comparison? The second issue is that I am not sure if the five metrics are comparable in an Apple-to-Apple manner, and whether it is fair to compare them directly. Based on the econometric specifications, Benchmark, Max, Min, and Wet Bulb seem to be comparable with the other two metrics. As a researcher who also works on heat exposure, I prefer the band specification over averages and counts, unless the data quality does not allow for this. Therefore, it is not usually an either-or question.

Is a high goodness of fit indicative of a better heat metric model? The third issue is that I am uncertain whether a higher goodness of fit indicates a better heat metric model. A higher goodness of fit metric explains the data variation better but does not necessarily capture the underlying mechanisms more effectively. I would imagine that some data-driven approaches, like Lasso and neural networks, would yield a much better goodness of fit, but they may not be preferred in research. The discussion regarding this issue is crucial to justify the methodology.

2.Connecting the theoretical channel to empirical results and interpretation of key parameters.

The three theoretical channels that the paper proposes and the empirical pattern do not seem closely tied. For example, from “Exposure to cooler temperatures that contribute to lower risk” to show the prolonged heat exposure to heat strain seems quite speculative to me. I am also not sure how you could test for the “Maternal behaviour” channel without data on factors such as labor participation and the performance of mothers.

My interpretation of Figure 4 would be quite different: overall null effect on heat exposure from the 1st and 2nd trimesters; substantial and close-to-linear effects for the 3rd trimester. I would dig more into the heterogeneity across trimesters.

3.Should one category being omitted in this research design?

Could you elaborate on why it would be necessary to omit one category for the band specifications? In the literature, this is commonly done because the total sum of each band is consistent for everyone, such as 365 days or 24 hours. However, in your case, pregnancy duration clearly varies for each baby. Unless you account for this in X_i, it seems there might be a missing effect from the overall duration of pregnancy.

4.The use of fixed effect model

By including the location-month-gender FE, the model is essentially comparing boys (girls) born in the same month and location. I would expect these fixed effects to absorb much variation that you would like to use for identification. Can you explain, e.g., using an example, what you are comparing conditional on these fixed effects?

Also, can your data track the location of the mother over the pregnancy cycle? Or are you assuming the location of the mother does not change? It seems to be an important assumption to explicitly discuss for the fixed effect model to control for omitted variable bias, such as potential sorting on place of residence and timing of conception.

5.Sample period.

Page 15, lines 298-301, the paper states: the administrative data is extracted from January 1994 to 31 December 2014, but the regression sample is from March 2000 to September 2009. I did not understand the justification behind this sample period selection.

Minor comments

1.Page 11, line 213: The font style of channel “Maternal behavioral” should be the same as the previous two channels.  

2.Figure 4, 7, 8: I recommend the authors consider breaking the coefficient for each trimester into a separate subfigure. The current deployment of the coefficient is quite messy and hard to interpret.  

3.Page 23, line 446, equation (1): I believe the usage of subscript itj for preterm_{itj} might be inappropriate as it could confuse people into thinking the data is individual-level panel data. For each baby i, there is no variation at t and j. I suggest just using preterm_{i} or preterm_{i(tj)}.  

4.Why does Figure 1 use the data from 2020-2023 while the main regression uses 2000 to 2009?  

5.In the first sentence of the abstract: “The impact of prenatal heat exposure on birth outcomes is well-established, but what is it about heat that affects prenatal development?” It is not clear to me why heat that affects prenatal development has to do with the main goal of the paper.  

6.Adding citation if you would like to speak more to the literature on environmental economics methodology literature.

Ghanem, D., & Smith, A. (2021). What are the benefits of high-frequency data for fixed effects panel models?. Journal of the Association of Environmental and Resource Economists, 8(2), 199-234.

Reviewer #2: I am very grateful for the opportunity to review the paper titled “The Relationship Between Prenatal Heat Exposure and Birth Outcomes: How Much Does the Heat Metric Matter?” I truly appreciate this opportunity.

I am still a graduate student, learning the process of reviewing. The length of my review and/or any missing information or sections may not meet your expectations, but please do let me know how I can improve.

I apologize again for the delay. Thank you for providing the extension.

The authors of this paper seek to answer the question of how heat exposure affects preterm birth by influencing fetal development through various pathways. While explaining their choice of heat metrics, the authors clearly establish that metrics appropriate for tropical climates may not be suitable elsewhere; for example, minimum temperature is not a reliable proxy for maximum temperature. This highlights the importance of carefully considering which aspect of heat exposure is being measured.

The conceptual framework, detailing how heat exposure may affect the fetus through heat stress, heat strain, and changes in maternal behavior, is clearly articulated. The paper also considers which specific components of heat, such as air temperature, humidity, or other environmental contributors, are most relevant for prenatal development.

The paper makes two main contributions. First, it quantifies five different ways of measuring heat exposures. Second, it evaluates which of these measurement choices matters most for preterm birth. The findings using the preferred heat exposure metric suggest that incorporating both daily maximum and minimum lead preterm birth, which adds to understanding that both moderate and extreme heat affect fetal development.

General Comments:

1. Good work

2. As the non-Aboriginal population tends to live in arid areas, which are more adapted to heat exposure, this may explain why they are not as strongly affected, as shown in Appendix E1, Column 1, and Appendix E2, Column 3.

3. Not sure if an example of animal studies will be good in this study!

4. Can you add max temperature bins ranges as well in a subsection of Max and Min, line 355

Clarification Comments:

5. Lines 283–284 mention that the sample includes babies born between March 2000 and September 2009, while lines 298–300 refer to birth records from the year 2017. Although a limited sample from March 2000 to September 2009 is acknowledged, line 302 states that the data analysis is prepared for the period 2023–2024, which creates confusion about the actual time frame of the data used. I think it would be helpful to have clarification on the final dataset, specifically, how many observations were ultimately used for the heat exposure analysis, how many birth records were matched, and other related details.

6. I wonder, on average, how many births are recorded per 50x55 km unit?

7. On a spatial scale, is there within-weather variation in a 50×55 km unit? I'm wondering how much of the variation occurs within that area

8. Also, do you think there may be heterogeneity by income levels as well? That is, does individuals have access to ways to mitigate heat exposure?

9. Do you observe how the main results differ between the Darwin area and the non-Darwin areas?

10. Could you show the minimum and maximum correlations across the entire study area, instead of only for Darwin, to better assess whether multicollinearity might be biasing the results?

11. I am curious, have you tried incorporating linear trends?

6. PLOS authors have the option to publish the peer review history of their article (what does this mean? ). If published, this will include your full peer review and any attached files.

**Do you want your identity to be public for this peer review?** For information about this choice, including consent withdrawal, please see our Privacy Policy .

Reviewer #1: No

Reviewer #2: **Yes: ** Prashikdivya Gajbhiye

---

## [Author Response · Author response to Decision Letter 1]

30 Jun 2025

Thank you very much for your feedback. We have carefully considered your comments - please see the attachment with full response to each comment

---

## [Decision Letter · Decision Letter 1]

3 Aug 2025

The relationship between prenatal heat exposure and birth outcomes: how much does the heat metric matter?

PONE-D-25-13859R1

Dear Dr. Doyle,

I have now heard back from both reviewers, who are satisfied with your revision. Reviewer #1 has a very minor comment on Table 2 and I am sure you will be able to correct that during the next phrase of the process. So at this point, I am pleased to inform you that your manuscript has been judged scientifically suitable for publication and will be formally accepted for publication once it meets all outstanding technical requirements.  I look forward to seeing the paper in print.

Kind regards,

Xinde James Ji

Academic Editor

PLOS ONE

Additional Editor Comments (optional):

I have now heard back from both reviewers, who are satisfied with your revision. Reviewer #1 has a very minor comment on Table 2 and I am sure you will be able to correct that during the next phrase of the process. I look forward to seeing the paper in print.

Reviewers' comments:

Reviewer's Responses to Questions

**Comments to the Author**

1. If the authors have adequately addressed your comments raised in a previous round of review and you feel that this manuscript is now acceptable for publication, you may indicate that here to bypass the “Comments to the Author” section, enter your conflict of interest statement in the “Confidential to Editor” section, and submit your "Accept" recommendation.

Reviewer #1: All comments have been addressed

Reviewer #2: All comments have been addressed

2. Is the manuscript technically sound, and do the data support the conclusions?

Reviewer #1: Partly

Reviewer #2: Yes

3. Has the statistical analysis been performed appropriately and rigorously? 

Reviewer #1: Yes

Reviewer #2: N/A

4. Have the authors made all data underlying the findings in their manuscript fully available?

Reviewer #1: No

Reviewer #2: No

5. Is the manuscript presented in an intelligible fashion and written in standard English?

Reviewer #1: Yes

Reviewer #2: Yes

6. Review Comments to the Author

Reviewer #1: All my comments in the previous review have been addressed or acknowledged in the paper.

I have only one minor comment on Table 2: Measures of model fit for each heat metric. For the first column, I suggest the authors report F-statistics instead of the p-value to be consistent with the latter two columns.

Reviewer #2: Dear Authors,

Thank you for incorporating the feedback and providing detailed explanations on all comments.

I appreciate your diligence and can see that you took my comments into careful consideration. I was surprised by the results for the sample of Darwin, the rest of the NT, and the linear trends.

Thank you also for adding the figure to the appendix.

The authors have responded adequately to the general and clarification comments. I believe the manuscript is ready to proceed to the next round.

7. PLOS authors have the option to publish the peer review history of their article (what does this mean? ). If published, this will include your full peer review and any attached files.

**Do you want your identity to be public for this peer review?** For information about this choice, including consent withdrawal, please see our Privacy Policy .

Reviewer #1: No

Reviewer #2: No

---

## [Editor Report · Acceptance letter]

PONE-D-25-13859R1

PLOS ONE

Dear Dr. Doyle,

I'm pleased to inform you that your manuscript has been deemed suitable for publication in PLOS ONE. Congratulations! Your manuscript is now being handed over to our production team.

Kind regards,

on behalf of

Dr. Xinde James Ji

Academic Editor

PLOS ONE